# Deriving an Analytical Expression for Core-collapse Supernovae Using Symbolic Regression

**Kaylee de Soto**
Center for Astrophysics | Harvard & Smithsonian
60 Garden Street
Cambridge, MA 02138
kaylee.desoto@cfa.harvard.edu

**V. Ashley Villar**
Center for Astrophysics | Harvard & Smithsonian
60 Garden Street
Cambridge, MA 02138
ashleyvillar@cfa.harvard.edu

## Abstract

Radiative transfer simulations of cosmic transients–the rapidly evolving terminal events of stars–are computationally expensive, making Bayesian inference infeasible on even a single events. Yet, astronomical surveys have discovered tens-of-thousands of these events. In this work, we use symbolic regression to derive an analytic expression for the luminosity of the most common core-collapse supernova (the explosive death of a massive star) as a function of time and physical parameters–an analytical expression for these events has eluded the literature for a century. This expression is trained from a set of simulated bolometric light curves (measured luminosity as a function of time) generated from six input physical parameters. We find that a single analytic expression can reproduce $\sim$70% of light curves in our dataset with less than $\sim$7.5% fractional error; we additionally present a small set of analytical expressions to reproduce the full set of light curves. By deriving an analytic relation between physical parameters and light curve luminosities, we create an interpretable parametric model and emulate the more expensive simulator. This work demonstrates promising preliminary results for future efforts to build interpretable emulators within time-domain astrophysics.

## 1 Introduction

Time-domain astrophysics is the study of cosmic events which evolve on hours to days timescales, including the explosive deaths of stars known as supernovae (SNe). SNe provide us a unique high-energy laboratory, the opportunity to study the creation of heavy elements in the universe, and direct tests of stellar evolution models. The Vera C. Rubin observatory [6] is expected to detect $\sim$1,000,000 SNe every year, providing unprecendently large datasets as well as a need for computationally efficient alternatives to radiative transfer models. In particular, we can use neural networks to map SN observables to physical properties, which would sidestep direct physical modeling, but these "black box" models are uninterpretable. Recent efforts have been made to develop physically interpretable machine learning models in the astrophysical literature, including group-invariant neural networks [4], physics-informed neural networks [1], and symbolic learning with inductive biases [3].

Core-collapse SNe are the explosive deaths of massive stars. We observe the light of these explosion as multi-variate time series – tracking photon flux as a function of wavelength and time. This is

37th Conference on Neural Information Processing Systems (NeurIPS 2023).

| Parameter Name | Units | Parameter Symbol | Grid Values |
|---|---|---|---|
| Progenitor Mass | $M_\odot$ | $M$ | 10, 12, 14, 16, 18 |
| Explosion Energy | $\times 10^{51}$ ergs | $E$ | 0.5, 1, 1.5, 2, 2.5, 3, 3.5, 4, 4.5, 5 |
| Nickel Mass | $\times 10^{-2} M_\odot$ | $M_{\mathrm{Ni}}$ | 0.1, 1, 2, 4, 6, 8, 10, 20, 30 |
| Log Mass Loss Rate | $-\log_{10}[M_\odot/\mathrm{yr}]$ | $\dot{M}$ | 1, 1.5, 2, 2.5, 3, 3.5, 4, 4.5, 5 |
| Circumstellar Radius | $\times 10^{14}$ cm | $R$ | 1, 2, 4, 6, 8, 10 |
| Circumstellar Structure | unitless | $\beta$ | 0.5, 1, 1.5, 2, 2.5, 3, 3.5, 4, 4.5, 5 |

Table 1: The physical parameters used to generate the Type IIP SN light curves in our dataset. Note that we have adjusted units to yield an approximately uniform grid across each parameter, where each value is close to unity. We aim to express light curve fits as a function of these values and time.

known as the "light curve." The shape of a SN's light curve depends on the mechanism of explosion as well as properties intrinsic to the progenitor star, such as its mass and the mass fraction of radioactive material. In this work, we build an emulator for physical models of the light curves of Type II SNe, which are core-collapse SNe from massive stars, from the physical parameters that impact the explosion. Type IIP SNe (SNe IIP), a subset of Type II SNe, are particularly challenging when it comes to simple analytical expressions. The light curves features a characteristic "plateau", caused by the recombination of hydrogen in the outer envelope of the progenitor star. The hydrogen-rich SN ejecta initially has a high opacity due to the ionized hydrogen. However, as the ejecta cools, the opacity drops, allowing photons deeper within the ejecta to escape. Furthermore, recent observational studies have revealed the ubiquitous nature of circumstellar material around the exploding star (see, e.g., [5]). This material causes yet another "phase change" in the SN light curve: one region first dominated by the interaction of circumstellar material and the SN ejecta, one dominated by the hydrogen recombination, and one finally dominated by the radioactive decay of newly synthesized elements. Despite years of study for simple scaling laws relating the light curve properties to the SN properties for Type IIP SNe [11, 7], the astronomical community still lacks a simple analytical expression for Type IIP light curves as a function of fundamental stellar properties and time.

Searching for an analytical representation of Type IIP light curves, we turn to symbolic learning. Symbolic regression differs from other machine learning techniques in that it aims to optimize an *analytical* relation between the inputs and outputs. Symbolic learning has increasingly been utilized by the astronomical community to develop such expressions [8, 9, 13]; however, symbolic learning has not yet been explored for SNe. Here, we utilize symbolic regression using a novel two-step approach. First, we use symbolic regression to fit a single light curve solely as a function of time with nonphysical constants. We then assume that form for all light curves, and fit the constants of that form as functions of the physical parameters. This will allow us to directly generate approximate light curves from a set of physical parameters without the need for computationally intense simulations or integration. All code is made publicly available[1].

## 2 Data and Methodology

Our SN IIP models are obtained from [10], and are generated from a set of six physical parameters further detailed in Table 1 and Section 2.2. These physical parameters can be sorted into three categories: (1) progenitor properties (progenitor mass, nickel-56 mass); (2) SN properties (explosion energy); and (3) circumstellar material (CSM) properties (mass-loss rate, CSM radius and CSM density structure). The complete dataset includes 228,016 light curves, uniformly sampled across this grid.

We pre-process the models such that a single light curve consists of 2,000 luminosities, with even spacing of 0.1 days (see Figure 1b). Given the extreme dynamic range of the luminosities, we model the base-10 logarithm of these luminosity values. As shown in Figure 1a, the light curves include three main regions: a pre-plateau luminosity peak (driven by the circumstellar material), a plateau (driven by hydrogen recombination), and a post-plateau region (driven by radioactive $^{56}$Ni decay). These regions are indicated in Figure 1a. A successful fit should properly model all three regions, which each provide unique physical insights about the progenitor star and SN.

---

[1]`https://github.com/kdesoto-astro/iip-symbolic`

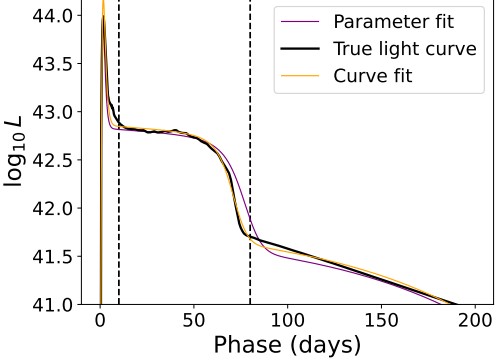

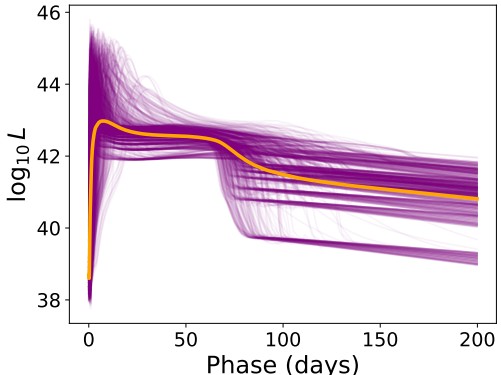

(a) A single light curve versus the modeled curves using `curve_fit` (orange) and the physical-to-fit parameter relations (purple). The three distinct "regions" of a IIP light curve are delineated by the vertical dashed lines.

(b) An overlay of all the bolometric light curves in our dataset (purple), as well as the mean light curve (orange).

Figure 1: The light curves we are trying to fit using analytic expressions. Note how there are three main regions per light curve: an early peak, a plateau, and a post-plateau decline. The duration and brightness of each region varies by light curve as functions of the physical parameters.

To perform the symbolic regression, we use the PySR[2] Python package [2], which is built on a Julia backend. PySR uses a genetic algorithm to explore symbolic expression trees to iteratively insert or remove operators, variables, and constants from the analytic expression. The loss function of each expression is the mean squared error obtained from all input and predicted log-luminosity values. Each expression is also assigned a "complexity" value, which for our defined model is simply the summation of the number of operations, variables, and constants in the expression. PySR keeps a running list, called the "hall of fame", of the one expression per complexity value $\text{complexity}_i$ with the lowest loss value. Each of these most accurate expressions $expr_i$ is then assigned a "score" given by:

$$\text{score}_i = -\log \frac{\text{MSE}_i}{\text{MSE}_{i-1}} \Big[\text{complexity}_i - \text{complexity}_{i-1}\Big]^{-1} \tag{1}$$

In practice, this score measures how much the extra degree of complexity contributes to reducing the mean squared error of the fit. The "best" expression is chosen as the expression within the highest score among the hall of fame expressions with a loss within 1.5 times the lowest loss on the list. This heuristic is the default used in PySR and ensures too much accuracy is not lost in search of a simple analytic expression.

## 2.1 General Form Derivation

Before attempting to use our physical parameters to fit all light curves simultaneously, we first establish a general form for each light curve as a function of time. We do this by choosing a "characteristic" IIP light curve (with clear peak, plateau, and fall) and running symbolic regression on that light curve with only the time steps as inputs. To enforce the three-part structure of the light curve, we enforce the inclusion of two sigmoid "transitions". Following training with PySR, this yielded the following best expression:

$$F(t) = C_1 + C_2 C_3^t + C_4 \sigma(C_5 + C_6^t) + (C_7 + C_8 t)\big[1 - \sigma(C_9 t)\big] \tag{2}$$

The first two terms ($C_1$ through $C_3$) represent the log-luminosity of the plateau region and thus centers the light curve vertically. The next term and associated sigmoid ($C_4$ through $C_6$) determine the plateau/post-plateau transition, and set the intensity of the post-plateau drop-off. The last term and sigmoid ($C_7$ through $C_9$) represent the pre-plateau luminosity peak, with peaks close to time zero. We demonstrate how each parameter impacts the light curve shape in Figure 2. In Figure 1a we see that this expression captures the light curve shape quite well, and we thus we use this functional form to fit the full grid.

---

[2]https://github.com/MilesCranmer/PySR

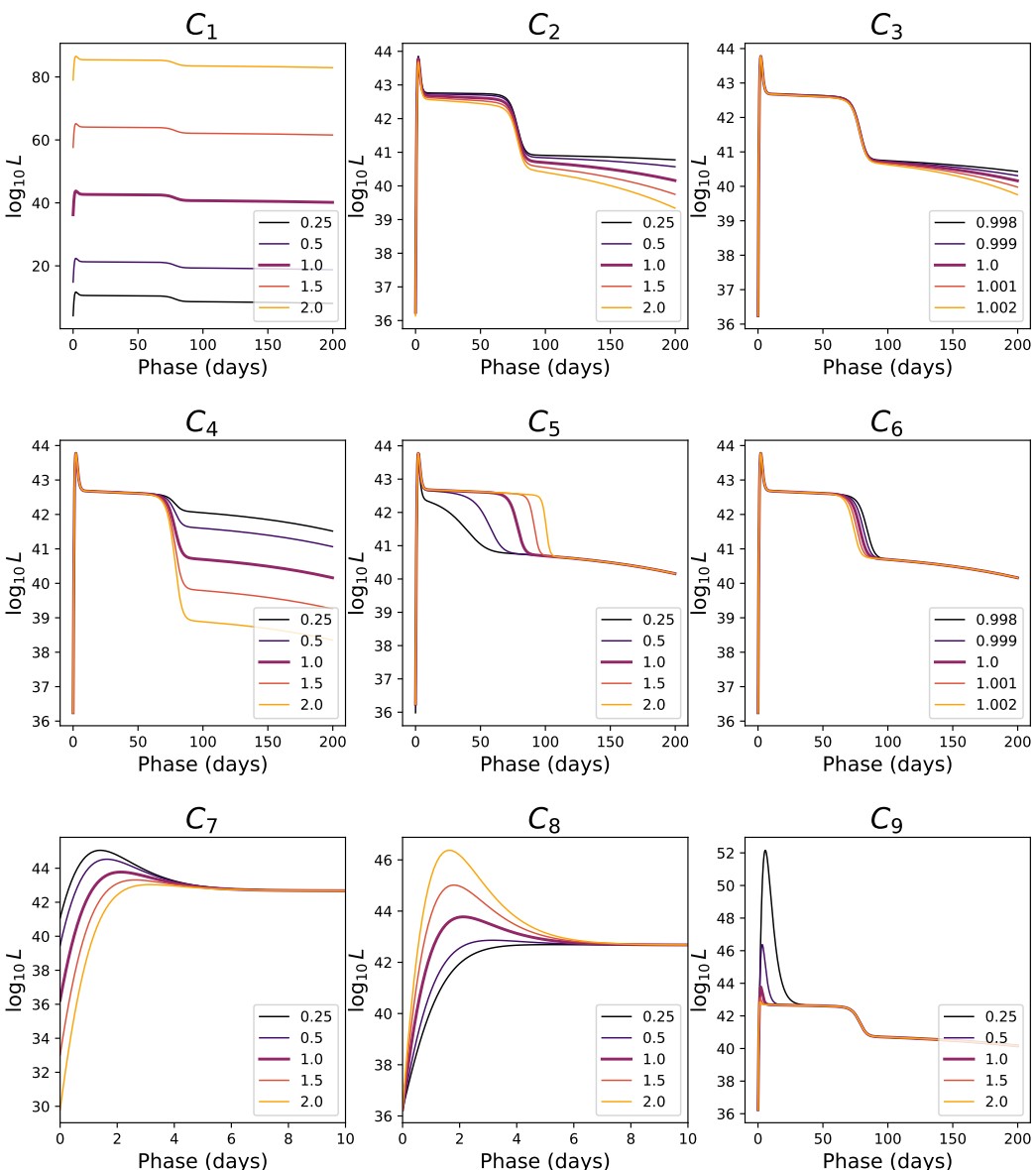

Figure 2: Demonstration of how each fit parameter in Equation (2) impacts the shape of the light curve. Note how the first parameter affects overall luminosity, $C_2$ through $C_4$ affect the post-plateau slope and amplitude, respectively, and $C_5$ and $C_6$ adjust the plateau duration and rapid decline, respectively, The remaining parameters affect the shape of the pre-plateau luminosity peak.

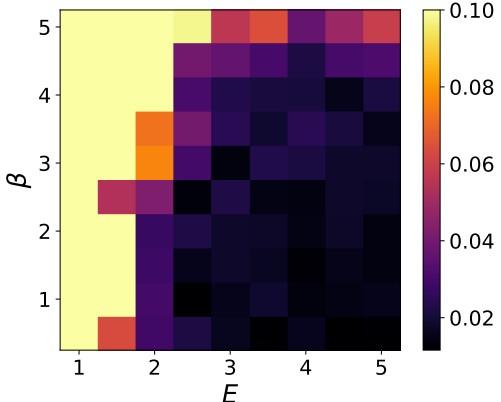

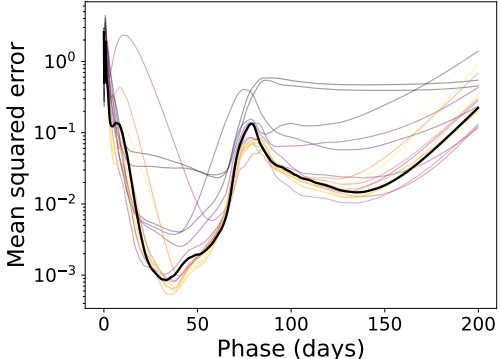

(a) Mean squared error of light curve fits (using `curve_fit`) for each value of explosion energy and circumstellar structure. Note that low energy light curves are much more poorly fit compared to high energy light curves. This indicates that a different general form is needed for those samples. The remaining physical parameters lacked clear delineations between high and low fit MSEs.

(b) Mean squared error over time of all the light curves fitted using the optimized physical parameter relations (in black). The MSE peaks pre-plateau, where the luminosity rapidly increases, and where the plateau ends around a phase of 80 days. The MSE values using alternate equations of increasing complexity are also shown, in increasingly warm colors.

Figure 3: Mean squared error (MSE) analyses of the light curves in our dataset using (a) direct curve fitting to our functional form and (b) our fit-to-physical parameter relations.

We then use Python's `curve_fit` package [12] with 5,000 maximum iterations to fit a random set of 1,000 light curves to the above general, yielding nine constants for each curve. We also save the mean-squared error (EMSE) for each fit, in units of log ergs per second, squared. From visual inspection, we determine that a MSE below 0.2 indicates an accurate fit, whereas a MSE below 0.5 indicates an adequate fit. These cutoffs correspond to absolute errors that are about 5% and 7.5% of the light curve amplitude, respectively, as the average per-light-curve standard deviation is $\sim 3$ log ergs per second. Approximately 35% of our light curves are fit with MSE $< 0.2$, whereas another 35% are fit with MSE $< 0.5$. As can be seen in Figure 3a, there is a clear cut in explosion energy $E$ and circumstellar structure $\beta$ that divides well fit and poorly fit light curves. In Figure 3 of [10], we see that low energy explosions lack the sharp luminosity peak present in higher energy explosions. Therefore, our functional form for IIP light curves is not ideal for these samples.

## 2.2 Relating Physical Parameters to Fit Constants

The SN IIP light curve models were generated from a set of six physical parameters: progenitor mass ($M$), explosion energy ($E$), $^{56}$Ni mass ($M_{Ni}$), mass loss rate ($\dot{M}$), circumstellar matter radius ($R$), and circumstellar structure parameter ($\beta$). The final parameter, $\beta$, defines the geometry of the circumstellar material such that the density of the material $\rho(r) \propto r^{-\beta}$, where $r$ is the distance of the material to the center of the progenitor star. These parameters were evenly sampled from a grid of possible values, shown in Table 1. Using these physical parameters as inputs and the above nine fit constants as outputs, we attempt to model a relation between the two sets of constants using symbolic regression.

In theory, the peak, plateau, and post-plateau regions are impacted by distinct subsets of the physical parameters. Therefore, in an attempt to limit each fit parameter to depend on fewer physical parameters, we increase the complexity of each variable from 1 to 2. This makes incorporating physical parameters twice as expensive as constants when optimizing best-fit equations, effectively distilling the most impactful physical parameters in each fit parameter's expression.

The resulting optimized expressions for the fit constants are as follows:

$$C_1 = 0.3948 \times \ln(\beta M) + 41.3792$$

$$C_2 = -\frac{2.2425 \times 10^{-3}}{M_{\mathrm{Ni}}^{\beta} + 0.0238}$$

$$C_3 = 0.0741 \times M^{\beta} \times \left(\frac{M_{\mathrm{Ni}}}{\beta - 1.1595} + 1.4456\right) + 1.0117$$

$$C_4 = 0.8680 - 0.4692\ln(M + \beta M_{\mathrm{Ni}}^{-1})$$

$$C_5 = \beta + \ln(M_{\mathrm{Ni}} + 0.0038) - \frac{22.8507}{\ln(M)}$$

$$C_6 = 1.0374 \frac{\ln(M_{\mathrm{Ni}})}{M}^{0.0075}$$

$$C_7 = 0.6181(E - \dot{M} + \ln R) - 12.8878 + \frac{0.8079}{10.8300 - M}$$

$$C_8 = \frac{\beta(5.1353E - R) + 31.5319}{M - 3.8763}$$

$$C_9 = M^{-1}\left[(287.6550M^{-2.2044} + \beta) \times (E - \ln(\dot{M} + R)) + 4.2604\right] \tag{3}$$

We compare our results to physical intuition. In particular, we can cross-reference Equation 2 with results from [10]. Starting with the post-plateau region, we see that $C_2$, $C_3$, and $C_4$ are all heavily impacted by the nickel-56 mass. This is expected: the light curve is primarily powered by nickel-56 decay following the recombination-powered plateau. On the other hand, we see from Figure 3 of [10] that the circumstellar radius primarily affects the luminosity peak; this is consistent with the fact that $R$ only appears in $C_7$, $C_8$, and $C_9$.

Note that inaccuracies in the `curve_fit` results propagate as inaccuracies in these expressions, so we expect additional noise when compared to the non-physical analytical models. Longer training time could potentially favor more accurate higher-order expressions, but we leave this exploration to future work.

The mean squared error of the fits across all light curves is 0.052, which corresponds to a fractional error of $\sim 7.5\%$. We plot the MSE as a function of time step averaged across all light curves in Figure 3b, highlighting that our model performs less well when fitting the rapid luminosity rise around phase zero, as well as the transition between plateau and drop-off. We also compare it to the MSE over time for expressions from complexity zero to twenty (the approximate complexity of our best expressions), indicating that we start seeing diminishing returns past complexity $\sim 7$ per expression.

## 3 Conclusions

In this work, we have shown that Type IIP SN light curves can be reasonably approximated by analytical expressions of physical parameters, uniquely determined via symbolic regression. Such an expression has eluded the literature for nearly a century. Here, our unique two-step approach allows us to first constrain a generic light curve form, and then find this form as a function of light curve parameters. Future work will focus on improving the maps between physical and fit parameters through longer training, thus ideally reducing the MSE at both very early times and during the plateau fall-off.

This simple analytical form allows us to build an interpretable emulator for Type IIP light curve models, enabling studies of thousands of events without the need for expensive radiative transfer simulations. Application of such a model to a large sample of observed Type IIP SNe will allows us to constrain the progenitors of the most common type of core-collapse SN.

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
