# OpenReview forum: "Representing Core-collapse Supernova Light Curves Analytically with Symbolic Regression"
_NeurIPS.cc/2023/Workshop/AI4Science — NeurIPS2023-AI4Science Poster_

### Official Review · Reviewer_PtHu · 2023-10-15
**Symbolic regression for Type IIP SN light curves**

**Rating:** 6
**Confidence:** 4

**Review:**

Symbolic regression is an approach that has been attracting research interests from the machine learning community. In this work, the authors attempt to derive a symbolic expression to explain Type IIP SN light curves. The experiment involves a generated dataset (sampled from a grid of possible values) and PySR.

The symbolic regression problem discussed in this study involve a phenomena that is presumably a hidden law rather than a known physics law (e.g, Kepler's equation), thus the reviewer finds the discussion interesting. This work also attempts to interpret the resulting symbolic expression for estimated constants in Eq. (3), which seems also interesting for the community, especially for symbolic regression researchers who are not physics experts.

Additional comments/questions:
- What does IIP stand for?
- What is the difference between SN vs. SNe?
- "Python’s curve_fit package" should be referenced (at least linked to the codebase)
- The plots (Figs. 3 and 4) seem too large. Those can be displayed horizontally
- "The “best” expression is chosen as the expression within the highest score among the hall of fame expressions with a loss within 1.5 times the lowest loss on the list." This seems very heuristic. Is there any justification?
- What is the unit of light curve? Since the range of the true values is not clear, it is difficult for the reviewer to assess how important that MSE achieved < 0.02 or 0.05.
- "35% of our light curve sample" -> "35% of our light curve samples"
- Last but not least, is the resulting F with estimated constants in Eq (3) really interpretable? The resulting F seems very complex

---

### Meta-Review · Area_Chair_ZdeQ · 2023-10-27

**Recommendation:** Accept (Poster)
**Confidence:** 3

**Metareview:**

This work uses symbolic regression to derive an analytic expression for the luminosity of the most common core-collapse supernova.
The symbolic regression problem discussed in this study involves a phenomenon that is presumably a hidden law rather than a known physics law (e.g., Kepler's equation), which motivates the reviewer's interest.

Given the practical relevance highlighted by the referee for both the workshop and also for the SR community, I recommend acceptance for this work. However, I urge the authors to incorporate the suggestions from the referee for the camera-ready version of the manuscript, as I agree with all of them.